# Resource partitioning among bat species in Peninsular Malaysia rice fields

Nur-Izzati Abdullah[1,2], Nurul-Ain Elias[3], Nobuhito Ohte[1] and Christian Vincenot[2,4]

[1] Department of Social Informatics, Kyoto University, Kyoto, Japan
[2] Island Bat Research Group, Kyoto, Japan
[3] School of Biological Sciences, Universiti Sains Malaysia, Gelugor, Penang, Malaysia
[4] Faculty of Science, Technology and Medicine, University of Luxembourg, Esch-sur-Alzette, Luxembourg

## ABSTRACT

Resource partitioning among tropical bats in agricultural areas of Peninsular Malaysia remains unclear. This study was conducted to evaluate resource partitioning among bats by examining their fecal samples. The main bat species sampled included: *Rhinolophus coelophyllus*, *Rhinolophus malayanus*, *Rhinolophus pusillus*, *Rhinolophus refulgens*, *Taphozous melanopogon* and *Hipposideros larvatus*. Two harp traps were set at different elevations on a hilltop (Gunung Keriang) and two high nets were used in neighboring rice fields at three sites, for three consecutive nights per sampling from April 2021 to February 2022. A total of 301 bats and 1,505 pellets were analyzed using a conventional approach which examined the fecal sample under the microscope. All of the bat species within the study had insects from the order Coleoptera, Lepidoptera, Diptera and Hemiptera in their diet. Larger bats exhibited a greater variety of prey consumption. Male individuals were observed to be generalists while female individuals were specialists, particularly during pregnancy and lactating reproductive stages. Bat species and insect order had a significant impact on the percentage fragment frequency of the insects consumed. *Rhinolophus coelophyllus* specialized in feeding on Coleoptera and Diptera, *H. larvatus* fed on Coleoptera, *R. malayanus* fed on Hemiptera, *R. pusillus* and *T. melanopogon* fed on Lepidoptera. Future molecular analysis can be carried out to further identify the insect pests consumed by these bats up to species level. These findings enhance our understanding of bats' ecological roles in agricultural landscapes and contribute to conservation and pest management strategies.

## INTRODUCTION

Bats provide us with important ecosystem services such as pollination, seed dispersal and regulating insect populations. Due to various preferences in dietary habits, bats consume a different variety of foods such as fruits, nectar and insects. Insectivorous bats play a vital role in controlling insects over a large-scale as most bats consume more than 50% of their body weight per night (*Tuttle, 1988*; *Kunz, Whitaker & Wadanoli, 1995*). The presence of insectivorous bats in agricultural areas, especially in rice fields, contribute to consuming insect pests that prey on crops. Nowadays, most farmers depend on chemical approaches

Corresponding authors
Christian Vincenot,
christian@vincenot.biz
Nurul-Ain Elias,
nurulain.elias@usm.my

(usage of pesticides) to control the insect pests in agricultural areas for instant results and greater crop benefits. However, in the long-term, this method will demonstrate a negative impact on the environment and indirectly increase the chemical resistance among insect pests.

Stem borers (order Lepidoptera), leafhoppers (order Hemiptera), planthoppers (order Hemiptera), and locusts (order Orthoptera) are the most common insect pests in rice fields (*Maisarah et al., 2015*). A previous study also recorded that the bearded tomb bat, *Taphozous melanopogon*, consumes a few species of insect pests in the surrounding rice fields at Gunung Keriang (*Nur-Izzati & Nurul-Ain, 2019*). Gunung Keriang (217 m above sea level) is a sole limestone hill that rises from the middle of an area of rice fields. It is a recreational area under the administration of Kedah State Government. Gunung Keriang is mostly visited by local residents for activities such as hiking and other outdoor pursuits. There is also calcite crystal extraction activity from several entrances of cave in Gunung Keriang. There is currently no specific legal measures or dedicated conservation efforts focused on bats, other cave-dwellers or the protection of their habitat in this location. However, it is crucial to consider the conservation status of bats and their habitats to implement appropriate conservation strategies and ensure the long-term survival of these flying mammals.

This hill comprises of various sizes of caves that provide bats with the stable thermoregulation and roosting space. Bat colonies that roosts in the caves at Gunung Keriang contributes to the large-scale depletion of insect pests in the rice fields surrounding the hill. There have been a few studies reported on bats at Gunung Keriang for the past 37 years. A study by *Hill, Zubaid & Davison (1985)* found a new record of *Hipposideros lekaguli* in Peninsular Malaysia. Since then, no study has been conducted to discover more bat species that can potentially aid in reducing insect pests in the rice fields surrounding Gunung Keriang.

Rice is a staple food consumed by the majority of Malaysians. To reduce the dependence on pesticides or insecticides, biological controls are a good option to be implemented in agricultural areas since this method has no adverse effects to the environment. For example, using natural predators, such as barn owls, which are introduced to rice field areas to reduce rat populations (*Hafidzi & Naim, 2003*). Malaysia is known to be a home to a huge number of bat species in the tropical region. The contribution from bats in controlling insect pests in agricultural areas, especially in rice fields, are mostly documented in the United States (*McCracken, Westbrook & Brown, 2010*; *Boyles et al., 2011*), some parts of Canada (*Clare et al., 2009*), Europe (*Puig-Montserrat et al., 2015*) and Madagascar (*Kemp et al., 2019*). In Southeast Asia, Thailand (*Leelapaibul, 2003*; *Leelapaibul, Bumrungsri & Pattanawiboon, 2005*; *Wanger et al., 2014*) has the highest reported studies of bats in rice field areas. However, the situation still remains unclear in Malaysia and other Southeast Asian countries. Studies on bats in Malaysian rice field areas are still lacking and more information is needed to determine their contribution to the ecosystem. We acknowledge the use of the preprint 'Bat community response to insect abundance in relation to rice phenology in Peninsular Malaysia' (*Nur-Izzati et al., 2023*), available on Authorea, which
provided valuable insights into the insect abundance and insect pests in the rice field area (https://doi.org/10.22541/au.168039363.37868412/v1).

The objective of this study is to evaluate resource partitioning among bats by examining their fecal samples. To date, there is a scarcity of data on the specific insect species consumed by bats in rice field areas. This study aims to shed light on the crucial role of bats in naturally controlling insect pest populations in rice fields. By emphasizing the significance of biological control in agricultural settings, it seeks to reduce the reliance on chemical methods for managing insect pests in the rice fields.

## METHODS AND MATERIALS

### Study area

Sampling was conducted in two different paddies' growing season (based on Muda Agricultural Development Authority, MADA's irrigation schedule):

Season 1—dry season (April 2021–September 2021)

Season 2—wet season (October 2021–February 2022).

The sites for the bat trapping in the rice fields and at Gunung Keriang (Fig. 1) were as follows:

(1) MADA A (6°10′59.2″N, 100°19′22.4″E, 3 m a.s.l.)—This site was located in a rice field area near residential areas (which have streetlights along the main road).

(2) MADA B (6°11′57.4″N, 100°19′50.0″E, 3 m a.s.l.)—This site was located in a rice field area near residential areas close to a small stream along the main road.

(3) MADA C (6°11′22.4″N, 100°20′30.2″E, 3 m a.s.l.)—This site was located in a rice field area near a stream that supplies water to the paddy plant and there were residential areas along the rice fields.

(4) GKRTOP (6°11′25″N, 100°19′56″E, 190 m a.s.l.)—This site was located at the highest part of the hill with dense vegetation and rocky terrain ascending towards the peak. Due to the frequent use of the trail by hikers, it provided an ideal location to set traps across the trail.

(5) GKRMID (6°11′30″N, 100°20′1″E, 70 m a.s.l.)—This site was located at the middle part of the hill and also has dense vegetation. As we ascend towards the middle elevation, we noticed the presence of relatively flat paths across the trail.

(6) GKRLOW (6°11′21″N, 100°20′5″E, 20 m a.s.l.)—This site was located at the lower part of the hill with dense vegetation and several small cave entrances. Observations indicated that only a few of these entrances were utilized by bats.

The trapping was conducted every month for three consecutive days in the rice field sites, MADA A, MADA B and MADA C, as well as at different elevations of Gunung Keriang, GKRTOP, GKRMID, and GKRLOW, starting from April 2021 until February 2022. There are five main phases of paddy growing, namely germination, vegetative, reproductive, ripening and harvesting. For the dry season, sample for the reproductive phase of paddy growth was excluded due to movement restriction. For the wet season, we recorded all the phases of paddy growth.

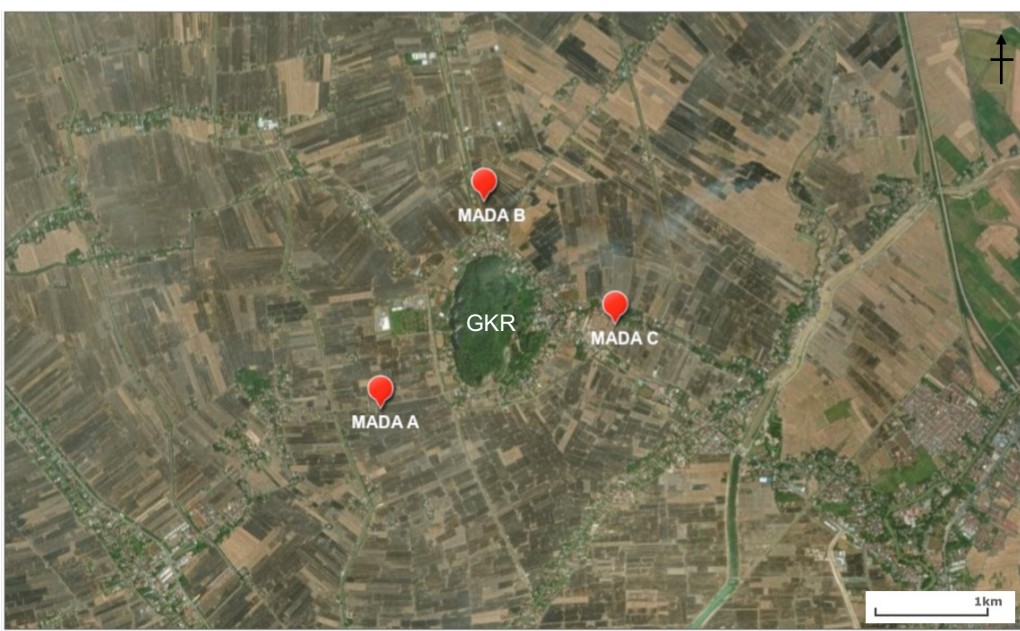

**Figure 1** **Location of MADA A, MADA B and MADA C that were chosen around Gunung Keriang (created using ArcGIS® software by Esri).**

## Bat trapping—fecal sample collection

In Malaysia, only several species of bats from the family Pteropodidae are listed as protected animals under the country's wildlife protection laws (Wildlife Conservation Act 2010) (*Government of Malaysia, 2010*). There are no specific legal provisions or regulations implemented on other bat species, or in place for their conservation in this region. We obtained the approval for our research protocol from Kyoto University (Inf-K23002). Insectivorous bats were captured using two harp traps at different elevations of the hill and two high nets were also used in the rice field areas at three sites (see subsection "Study Area"). High nets were set up at dusk (1900 h), were checked every 10-15 min and closed at 2200 h. Harp traps were set up at midnight (0000 h), were checked every hour and closed just after dawn (0700 h). Captured bats were weighed, sexed and identified based on their morphology (*Kingston, Lim & Zubaid, 2006*; *Francis, 2008*). Bats were kept in a cloth bag individually for less than four hours for fecal sample collection before they were released at the point of capture. We followed the protocol by *Kunz, Hodgkison & Weise (2009)* and *Kingston et al. (2021)* for capturing and handling bats in this study.

The conventional approach was used for dietary study with different purposes. Our major focus was on the conventional approach to understand overall resource and food partitioning among bats. Six bat species were chosen for this approach. Four bat species from the family Rhinolophidae were chosen due to the high number of individuals captured, which included *Rhinolophus coelophyllus*, *Rhinolophus malayanus*, *Rhinolophus pusillus* and *Rhinolophus refulgens*. The two additional bat species: *Taphozous melanopogon* from family Emballonuridae were chosen as the representatives of open space bat species

and *Hipposideros larvatus* was also chosen as a representative from family Hipposideridae. *Taphozous melanopogon* and *H. larvatus* did not have a sufficient sample size since they were not the most abundant bat species captured.

## Fecal analysis—conventional

The fecal pellets were dried at 80 ° C for 8-10 min to prevent the growth of moss and fungus in the feces that would disintegrate fecal samples (*Whitaker, McCracken & Siemers, 2009*). The dried fecal pellets were stored in 1.5 ml Eppendorf tubes together with naphthalene powder. For each species, 10 individuals were randomly chosen (five males and five females) for each paddy growing phase. For each individual, five of the fecal pellets were chosen and examined, which is considered sufficient to determine the diet of a single bat (*Whitaker, Neefus & Kunz, 1996*; *Leelapaibul, Bumrungsri & Pattanawiboon, 2005*). Each pellet was soaked and softened in a petri dish with 70% alcohol for 15 min (*Fenton et al., 1998*) and was dissected using insect needles and fine forceps. The fecal pellets were observed under a stereo-microscope and the prey items in the sample were identified by using a reference from a complete insect fragment collection (*Wilson & Claridge, 1991*). All the insects' parts that were observed under the microscope were identified up to their order level (*Wilson & Claridge, 1991*). The fragments observed were recorded on the fecal analysis datasheet. Photos insect fragments were taken for proper documentation and publication.

## Data analysis

The *landscapemetrics* package was used to conduct the landscape fragmentation analysis. For this analysis, we used maps from the OpenStreetMap (OSM) (http://www.openstreetmap.org/), which we imported into ArcGIS 1.2 for preprocessing purposes. Specifically, we extracted the layers 'forest cover' and 'agriculture land' layers, focusing on the 'rice paddy' category. These layers were converted into black-and-white raster maps, representing the presence or absence of the respective land types. Subsequently, the maps were then exported as raster files and imported into an R script, where the *landscapemetrics* package was used to calculate landscape-level fragmentation metrics. Using this package, we computed the metric 'number of patches' using the 'lsm_l_np()' function, which represents the number of patches formed by contiguous cells of a specific land cover type, such as forest and rice fields for hill side and agriculture land sites, respectively. These metrics provide a summary of the whole landscape, condensing it into one value (*McGarigal, Cushman & Ene, 2012*; *Hesselbarth et al., 2019*). The calculation of the metric incorporates all patches, resulting in a single numerical output. The patch count metric offers valuable insights into landscape fragmentation, connectivity and the distribution of land cover within the landscape.

Identified insect parts in the feces were calculated by using percentage fragment frequency (*McAney, 1991*) and percentage volume (*Whitaker, Neefus & Kunz, 1996*). Percentage fragment frequency was expressed by the number of occurrences of categories, then divided by total occurrences of all categories, multiplied by 100. As for percentage volume, a grid paper of 60 mm × 60 mm was used. The occupancy of the insect parts based on the grid paper's scale was measured. The percentage volume then was expressed by the

number of grids occupied by the order divided by the total number of grids, multiplied by 100. The Hardness index for the diet of each bat species was calculated using the bat's volumetric prey consumption (*Freeman, 1981*) and hardness value (HV) ranging from 5 (hardest) to 1 (softest) for each insect order (1 for Ephemeroptera, Isoptera, Trichoptera, Diptera; 2 for Odonata, Homoptera, and Lepidoptera; 3 for Orthoptera; 4 for Hemiptera, Hymenoptera; and 5 for Coleoptera). For bat species with multiple dietary descriptions, the index of hardness was averaged (*Ghazali & Dzeverin, 2013*).

The Shannon Index was calculated to determine the diversity of the diet intake within each species and also the bat diversity between sites. A dendrogram was plotted to visualize the similarity among the four bat species. The corrplot package was used to visualize the Chi-square test of independence, which determined whether there is a significant relationship between bat species and insect order. A mean comparison was performed using a two-way ANOVA to differentiate between bat species and insect order based on the percentage frequency of food consumed. Furthermore, a three-way ANOVA was conducted to examine the effects of bat species, different reproductive stages and insect order on the percentage frequency within the female individuals. All the statistical analyses were performed using R version 4.2.2 (*R Core Team, 2002*).

# RESULTS

Based on personal observations, we assumed that the sites on the hill (GKRTOP, GKRMID, and GKRLOW) were relatively similar to one another in terms of fragmentation and that same goes for the rice field sites (MADA A, MADA B and MADA C). The statistical analysis revealed that there was only one continuous patch present at both the hill and rice fields sites. Yet, when comparing the differences between the hill and rice field sites in terms of bat diversity by using the Shannon Index, we found that GKRTOP ($H = 2.21$) exhibited the highest bat diversity, followed by the GKRMID ($H = 2.07$) and GKRLOW ($H = 1.91$). Among the three identified sites on the rice fields, MADA B ($H = 0.93$) showed a higher bat diversity compared to MADA C ($H = 0.69$) and MADA A ($H = 0.26$). Hence, even though the analysis of patch density revealed that both the hill sites and rice field sites possess similar levels of fragmentation, the sites on the hill and rice fields exhibited different bat diversity.

A total of 301 individuals of bats with 1,505 pellets were analyzed for the dietary study. Overall, the studied species commonly have Coleoptera, Lepidoptera, Diptera and Hemiptera in their diet. Meanwhile, order Hymenoptera, Trichoptera, Odonata, Isoptera, Homoptera, Ephemeroptera and unidentified fragments were recorded in small percentages in a few of these species. 80% of the volume of the insects consumed were Coleoptera and Lepidoptera. Photos of insect fragments in the fecal sample were taken for the common insect order such as Coleoptera, Lepidoptera, Diptera and Hemiptera (Fig. 2).

Based on the percentage fragment frequency, *Rhinolophus coelophyllus* consumed insects from the order Coleoptera (74.29%), Lepidoptera (20.72%), Hemiptera (2.47%), Diptera (2.30%) and Trichoptera (0.22%) in the dry season. While in the wet season, the insect group preferences were still the same except for Trichoptera that were not found in their

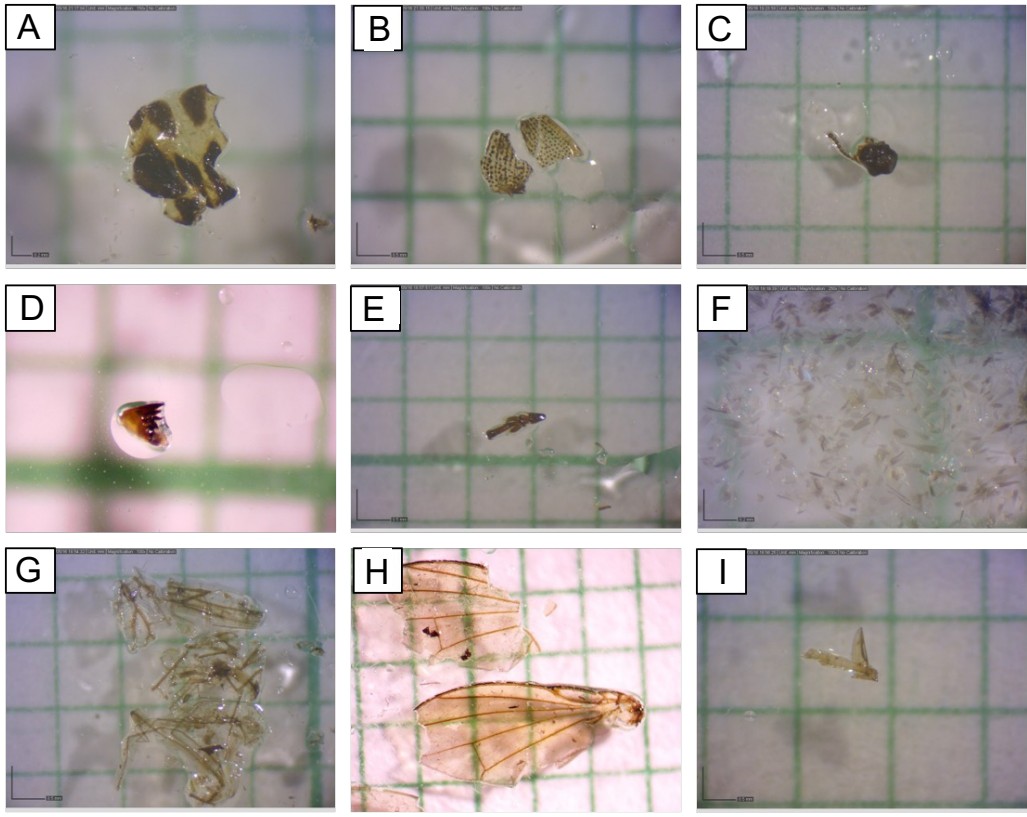

**Figure 2** **Common insect fragments found in fecal samples observed under microscope.** (A) Coleoptera (body), (B) Coleoptera (body), (C) Coleoptera (head), (D) Coleoptera (mouthpart), (E) Coleoptera (leg), (F) Lepidoptera (scales), (G) Diptera (wings), (H) Diptera (wings), and (I) Hemiptera (leg).

fecal samples (Fig. 3). However, based on the percentage volume, in the dry season, Coleoptera (92.73%) and Hemiptera (27.53%) were mostly found in the fecal samples of this species, whereas in the wet season, two major insect groups were Coleoptera (74.37%) and Lepidoptera (35.76%) (Fig. 4). Only male individuals of *Rhinolophus coelophyllus* consumed Trichoptera (0.28%) but none of female individuals recorded any fragments of Trichoptera. Male individuals also consumed a significant number of Hemiptera (26.35%) compared to females (0.20%). In the dry season, pregnant females of *R. coelophyllus* exclusively fed on Coleoptera, while non-reproductive females consume insects from Coleoptera and Lepidoptera. Lactating females consumed insects from the order Coleoptera, Lepidoptera, Diptera and Hemiptera. No post-lactating females were recorded in the dry season. In the wet season, post-lactating females consume insects from the orders Coleoptera, Lepidoptera and Diptera.

*Rhinolophus malayanus*, in the dry season consumed insects from Coleoptera (44.82%), Lepidoptera (33.70%), Hemiptera (10.08%), Diptera (7.67%), Odonata (1.24%) and Hymenoptera (0.48%) while in the wet season they consumed Coleoptera (62.47%), Lepidoptera (31.81%), Diptera (3.57%) and Hemiptera (2.15%) (Fig. 3). In the dry and

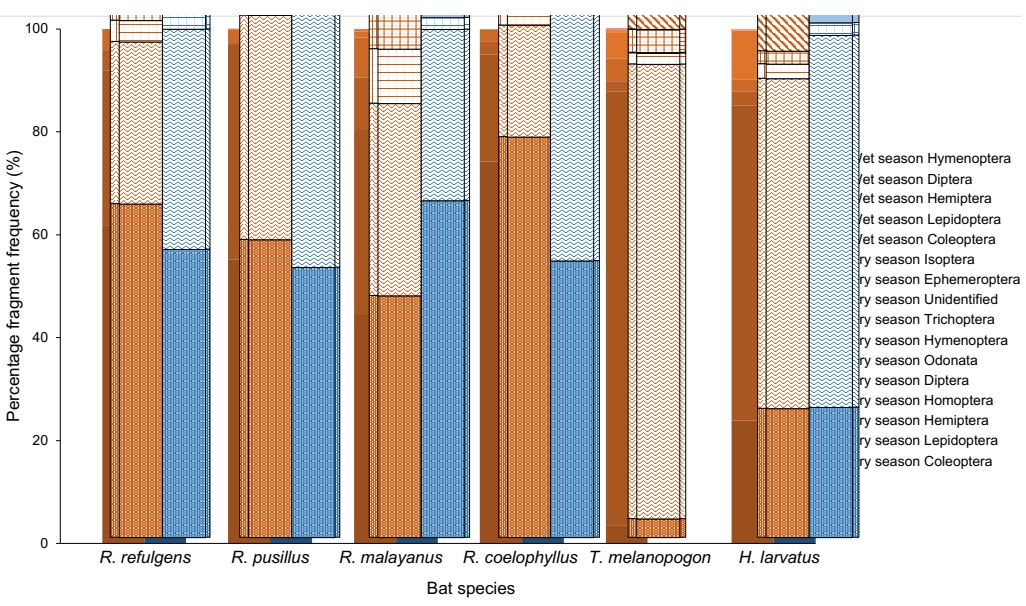

**Figure 3** Percentage fragment frequency of six bat species according to different insect order in the dry and wet season.

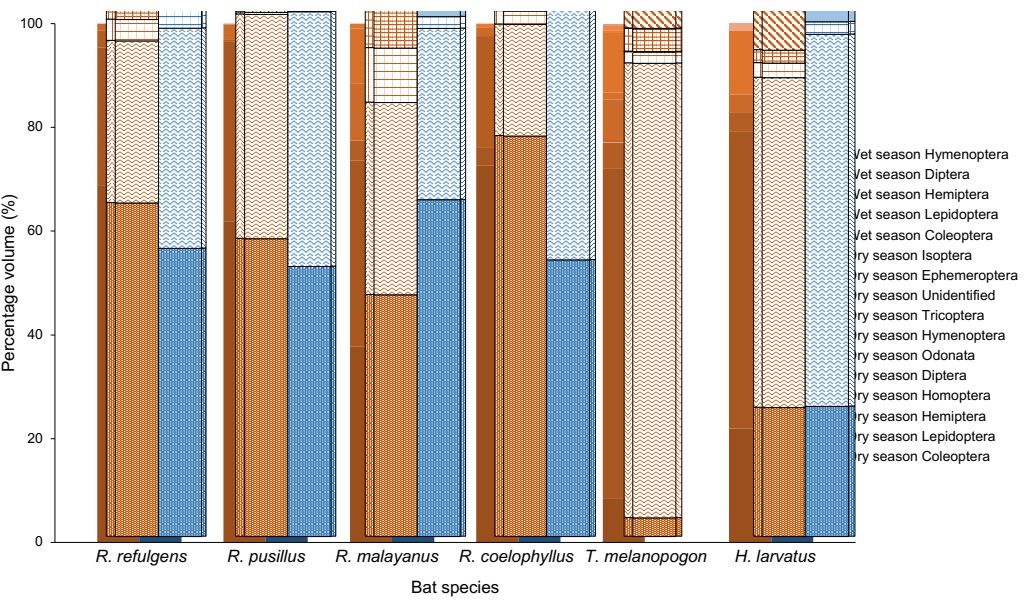

**Figure 4** Percentage volume of six bat species according to different insect order in the dry and wet season.

wet seasons, *R. malayanus* consumed a high volume of Coleoptera and Lepidoptera (Fig. 4). Only female individuals of *R. malayanus* consumed Hymenoptera (0.84%) and Odonata (2.15%). Males consumed a small amount of Hemiptera (2.83%) and Diptera (4.88%) compared to females (Hemiptera (17.55%) and Diptera (17.45%)). In the dry season,

pregnant females consumed only Coleoptera, while post-lactating females consumed a wide range of insects including Odonata and Hymenoptera but did not consume any Lepidoptera at all.

*Rhinolophus refulgens*, in the dry and wet seasons, consumed insects from the same insect order (Fig. 3). In the dry season they consumed Coleoptera (61.87%), Lepidoptera (30.07%), Diptera (4.08%) and Hemiptera (3.98%) while in the wet season they consumed Coleoptera (55.46%), Lepidoptera (42.39%), Hemiptera (1.10%) and Diptera (1.05%). In the dry and wet seasons, *R. refulgens* consumed a high volume of Coleoptera and Lepidoptera. For Diptera, even though in the dry season this species recorded a high percentage fragment frequency (4.08%) (in wet season 1.05%) but for volume percentage in the wet season they were present in a high number (12.02%) compared to the dry season (4.74%) (Fig. 4). Female individuals of *R. refulgens* consumed a small number of Hemiptera (0.50%) compared to male individuals (5.32%). For both dry and wet seasons, pregnant and lactating females prefered to consume Coleoptera and Diptera while post-lactating females consume Coleoptera, Diptera, Lepidoptera and Hemiptera. There were no differences in preference for dry and wet seasons.

*Rhinolophus pusillus*, in the dry and wet seasons consumed insects from the same insect order (Fig. 3). In the dry season, they consumed Coleoptera (55.23%), Lepidoptera (41.62%), Diptera (2.54%), Hemiptera (0.39%) and Hymenoptera (0.23%). In the wet season they consumed Coleoptera (50.10%), Lepidoptera (47.27%), Diptera (2.36%), Hemiptera (0.17%) and Hymenoptera (0.10%). In the dry and wet seasons, *R. pusillus* consumed a high volume of Coleoptera and Lepidoptera (Fig. 4). Both male and female individuals of *R. pusillus* were recorded to consume a small number of Hemiptera and Hymenoptera in their diet. All females of different reproductive stages consumed a large number of Coleoptera and Lepidoptera.

*Taphozous melanopogon* was only recorded in the dry season. In the dry season, they consumed insects from the order Lepidoptera (84.37%), Hymenoptera (5.09%), Diptera (4.21%), Coleoptera (3.44%), Hemiptera (2.05%), Unidentified (0.30%), Ephemeroptera (0.21%), Trichoptera (0.14%), Homoptera (0.11%) and Odonata (0.09%) (Fig. 3). The percentage volume of insects consumed was lower in Lepidoptera (65.15%) but double in Hymenoptera (11.78%) when compared to the percentage fragment frequency of consumption (Fig. 4). Male and female individuals were not much different in terms of insect order consumed. But within female individuals, Trichoptera and Ephemeroptera were not consumed by them and there was no preference shown in different reproductive stages of the female individuals as well.

*Hipposideros larvatus* in dry and wet season consumed insects from the same order. In the dry season, this bat species consumed insects from the order Lepidoptera (61.20%), Coleoptera (23.92%), Hymenoptera (9.44%), Hemiptera (2.72%), Diptera (2.39%) and Isoptera (0.34%). In the wet season, the percentage of the consumption of the insects was slightly different with Lepidoptera (69.03%), Coleoptera (24.14%), Diptera (2.36%), Hemiptera (2.34%), Hymenoptera (2.13%) and no insect fragments from the order of Isoptera were recorded in this season (Fig. 4). The percentage volume of the insects consumed did not differ much between the dry and wet seasons. Male and female individuals

consumed the same insect orders. Among the female individuals in different reproductive stages, pregnant female consumed a higher quantity of Lepidoptera compared to females in other reproductive stages.

Coleoptera and Lepidoptera fragments were found in large numbers in the fecal samples of four of these bat species. Since Coleoptera is a hard-bodied insect, elytra, leg, headpart and mouthpart are the most common parts that can be easily found in the fecal samples of insectivorous bats. However, for the soft-bodied insects, most of the identified body fragments were the leg, wing and scales (only for Lepidoptera for *Nilaparvata lugens* the stem borer). In this study, the prey hardness varied among the bat species with *R. malayanus* consuming the hardest prey while *T. melanopogon* consumed the softest prey.

Based on the Shannon index, *R. malayanus* ($H = 1.14$) had the highest dietary diversity, followed by *H. larvatus* ($H = 0.98$), *R. refulgens* ($H = 0.87$), *R. pusillus* ($H = 0.82$), *R. coelophyllus* ($H = 0.80$) and *T. melanopogon* ($H = 0.68$). *Taphozous melanopogon* were recorded to consume 10 different insect orders, *R. malayanus* and *H. larvatus* with six different insect orders while *R. pusillus* and *R. coelophyllus* consumed five insect orders and *R. refulgens* consumed four different insect orders. Although *T. melanopogon* had a diverse food intake compared to others, this bat species still possessed the lowest diversity index of the food consumed due to the high amount of Lepidoptera and small percentages for the other insect orders. *Taphozous melanopogon* and *H. larvatus* had a more diet than *Rhinolophus* species. Within the *Rhinolophus* species, *R. coelophyllus* and *R. refulgens* had the same food content while *R. malayanus* had the same dietary intake with these two species. *Rhinolophus pusillus* had a different food intake than *R. coelophyllus*, *R. refulgens* and *R. malayanus* (Fig. 5).

Based on the chi-Square test of independence, there was a significant relationship between the overall bat species and insect order consumed, $X^2(50) = 172.49$, (*p*-value $< 0.001$). The species of bats were highly dependent on the order of insects consumed. Based on the standardized residuals of the association between the bat species and insect order, there was a strong positive relationship between *R. coelophyllus* and *R. refulgens* with Coleoptera, *R. malayanus* with Hemiptera, Odonata and Diptera, *T. melanopogon* with Lepidoptera and Hymenoptera, *H. larvatus* with Hymenoptera and Lepidoptera (Fig. 6A). As shown in Fig. 6B, the cells most contributing to the Chi-square value were *R. coelophyllus*/Coleoptera (5.41%), *R. refulgens*/Coleoptera (3.51%), *R. malayanus*/Hemiptera (4.36%), *R. malayanus*/Lepidoptera (3.40%), *R. malayanus*/Odonata (3.39%), *R. pusillus*/Hemiptera (1.54%), *T. melanopogon*/Coleoptera (22.18%), *T. melanopogon*/Lepidoptera (17.24%), *H. larvatus*/Hymenoptera (6.42%) and *H.larvatus*/Coleoptera (5.51%). These cells contributed about 72.96% to the total Chi-square score and thus accounted for most of the difference between expected and observed values.

The different bat species did not have a significant effect on the percentage fragment frequency of food consumed by bats, $F_{5,1998} = 0.76$, $p = 0.58$. However, insect order had a significant effect on the percentage fragment frequency of the food consumed, $F_{10,1998} = 97.86$, $p < 0.001$. There was a statistically significant interaction between the effects of bat species and insect order on the percentage fragment frequency of food

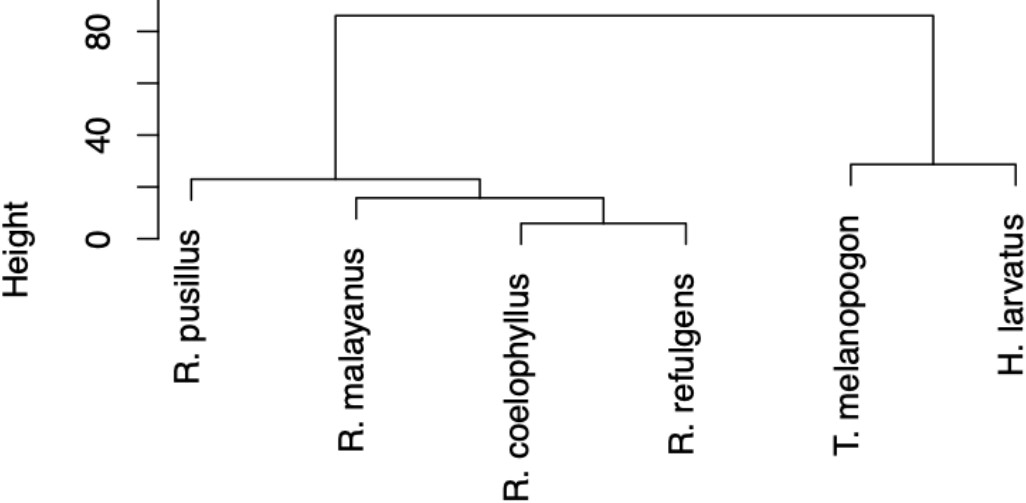

**Figure 5** **A dendrogram of the food partitioning of bat species.**

consumed, $F_{50,1998} = 13.34$, $p < 0.001$. Within the female individuals, the results revealed significant main effects of bat species ($F_{5,290} = 7.90$, $p < 0.001$), different reproductive stages ($F_{3,290} = 4.75$, $p = 0.003$) and insect order consumed ($F_{7,290} = 104.39$, $p < 0.001$), indicating that all of these factors significantly influenced the percentage fragment frequency of the food consumption of bats. None of the interactions between the three variables were statistically significant, $F_{27,290} = 1.07$, $p = 0.38$. We ran a separate one-way and two-way ANOVA to prove that which variables (bat species, reproductive stages or insect order) had an influence on the percentage fragment frequency. Based on the results, we can conclude that the different reproductive stages did not have any effect on the percentage fragment frequency and this variable had a huge impact on the interactions between the three variables.

## DISCUSSION

Similar levels of fragmentation were observed between the sites on the hill and in the rice fields. This similarity may be attributed to the dense vegetation that covers most of the hill, resulting in consistent fragmentation patterns across the sites. In the rice field landscape, there was a single distinct patch present in this land use type. Different sites on the hill and in the rice fields had an impact on the bat diversity. Even though the sites in the rice fields

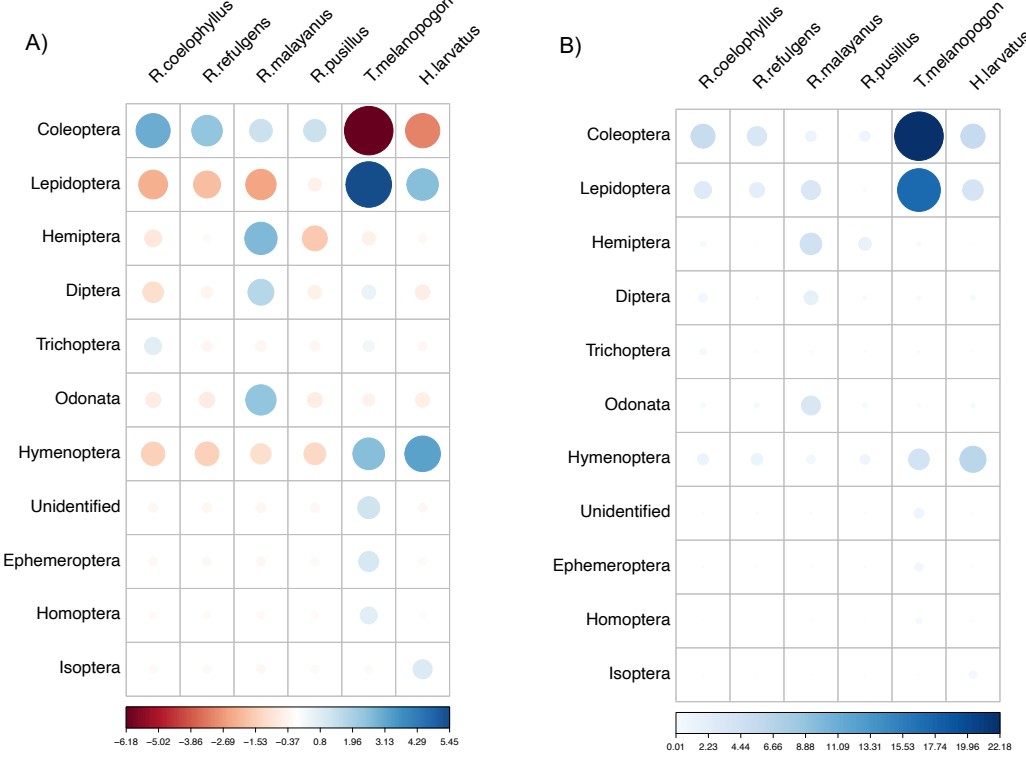

**Figure 6** **The visualization chi-square score using corrplot package.** (A) standardized residuals and (B) the dependency between bat species and insect Order. The color and size indicate the strength of the association with the darker blue and larger circle show the highest association.

exhibited lower bat diversity, we were able to capture *R. malayanus* and *H. larvatus* using high nets. This suggests that the hill serves as a roosting site for these species, while the rice field areas serve as their foraging sites. Although we did not conduct any radio tracking or mark recapture to further confirm this, the home ranges of these two species are assumed to fall within the vicinity of the hill, supporting our observation. *Kingston (2000)* found that cave-roosting bat species generally exhibit higher vagility compared to forest-roosting bat species, as they are capable of efficient commuting between caves to their foraging grounds. Radio-tracking studies have shown that cave-roosting Rhinolophidae and Hipposideridae can commute several kilometers during a single night (*Pavey, Grunwald & Neuweiler, 2001*; *Bontadina, Schofield & Naef-Daenzer, 2002*; *Struebig et al., 2008*). Additionally, these bats also showed a greater resilience to withstand the effects of habitat fragmentation (*Struebig et al., 2009*).

The diet of bats is greatly influenced by regional and temporal factors (*Whitaker, Neefus & Kunz, 1996*), as well as seasonal variations (*Leelapaibul, Bumrungsri & Pattanawiboon, 2005*; *Zhang et al., 2005*; *Liu et al., 2019*). In the tropics, with no obvious seasonal pattern as in temperate regions, the insect availability and abundance differ between the dry and wet season (*Kishimoto-Yamada & Itioka, 2015*; *Nurul-Ain, Rosli & Kingston, 2017*). It is also associated with the phase of paddy growth, due to the variations in insect species diversity

and quantities (*Maisarah et al., 2015*; *Liu et al., 2019*). In our study, Coleoptera served as the primary food source for the all bat species, albeit with varying percentage consumption during the dry and wet season. Interestingly, during the wet season, although insects from the order Homoptera were the most captured insect order (*Nur-Izzati et al., 2023*), they were not identified as the main food source for the bat species as no Homoptera fragments were found in the fecal samples. However, we observed that only *T. melanopogon* consumed a small amount of Homoptera during the dry season. It is possible that insects from the order Homoptera cater to the food preferences of other bat species.

In this study, Coleoptera was the most abundant prey found in all these six bat species followed by Lepidoptera, Diptera and Hemiptera. A preliminary diet analysis study of *T. melanopogon* (black-bearded tomb bat) in rice fields around Gunung Keriang revealed that their main insect consumption was from the orders Lepidoptera, Coleoptera and Diptera (*Nur-Izzati & Nurul-Ain, 2019*). Previous studies on the dietary composition of *T. melanopogon* conducted by *Zubaid (1990)* found that roosting individuals in caves were generalists, while those in forest exhibited more opportunistic feeding behavior. Similarly, *Srinivasulu & Srinivasulu (2005)* documented that *T. melanopogon* in forested areas primarily consumed Coleoptera and Lepidoptera in semi-urban ecosystems. In this study, *T. melanopogon* displayed a notable preference for Lepidoptera, along with a diverse range of insect prey. This suggests that *T. melanopogon* are opportunistic feeders, consuming insects encountered during foraging, consistent with *Zubaid*'s findings (*1990*). *Wei et al. (2006)* also observed that *T. melanopogon* consume larger insect prey and return to the roost later compared to other bat species, likely due to their long distance flight capabilities.

Lepidoptera, Coleoptera and Hymenoptera were found in the diet of *Hipposideros larvatus* and these insect orders were also recorded as prey for *Hipposideros armiger* (*Zubaid, 1988a*). Additionally, *Zubaid (1988b)* reported that *Hipposideros pomona*'s diet consisted of 90% of Odonata and Lepidoptera. Comparing their sizes, *H. larvatus* (FA = 51–67 mm, body weight = 15–23 g) is slightly smaller than *H. armiger* (FA = 85–103 mm, body weight = 44–67 g). The dietary habits of *H. armiger* suggest that this species not only catches insects in flight but also picks up insects from vegetation and surrounding objects, indicating an ability to hover (*Zubaid, 1988a*). The high proportion of insect fragments from the order Lepidoptera in the diet of *H. larvatus* suggests that this species primarily consumes fluttering insects present in the environment.

*Wei et al. (2006)* discovered that insects from Lepidoptera dominate the prey items in *R. pusillus*, which are also found in these four *Rhinolophus* bat species: *R. coelophyllus*, *R. malayanus*, *R. pusillus* and *R. refulgens*. However, in this study, these *Rhinolophus* species primarily consumed Coleoptera, followed by Lepidoptera. A study by *Mohamed, Sah & Nurul-Ain (2023)* in Peninsular Malaysia identified Coleoptera as the main order consumed by a *Rhinolophus* species, a result consistent with our study. The underlying reason for these differences could be that *Rhinolophus* species might have adapted to consume Coleoptera in the environment, as this insect order is the most abundant in the area (*Nur-Izzati et al., 2023*). Bats from Rhinolophidae also are not restricted in capturing winged insects, these bats also capture non-winged prey such as Chilopoda (centipedes), Dermaptera

(earwigs), Arachnida (spiders, ticks) and Mesostigmata (mites) (*Ahmim & Moali, 2013*; *Pavey, 2021*). The orders Odonata (dragonflies), Hymenoptera(wasps) and Diptera (bat fly) have been consumed by bats in small amounts. This is due to accidental consumption while grooming, catching prey or picked during flight (*Whitaker & Lawhead, 1992*; *Ahamad et al., 2013*; *Ramanantsalama et al., 2018*) or being consumed by foraging bats near artificial light sources at dusk (*Liu et al., 2019*).

During this study, we observed bats foraging at the streetlights nearby the rice fields. Streetlights also produce the 'dazzling effect', which refers to the behavior exhibited by insects when they are attracted to light sources. The insects become dazzled and immobilized as they approach the light, causing them to land on the ground and become an easy target to many predators, including bats. Lepidopterans such as moths often prefer artificial light such as street lights rather than the moon to travel back and forth (*Eisenbeis, 2006*). Coleoptera are also attracted to light (*Medeiros, Barghini & Vanin, 2017*). Open space bat species, such as *T. melanopogon* or *Miniopterus magnater*, tend to forage near streetlights (*Boonchuay & Bumrungsri, 2022*) due to their morphology, which enables them to hover around as the light attracts insects. The presence of insects swarming near the light indirectly attracts bats to streetlights. However, bats from the family Rhinolophidae and Hipposideridae do not hover around the streetlight due to their limitation in morphological characteristics (low wing loadings and low aspect ratio) (*Stone, Jones & Harris, 2012*).

*Feldhamer, Carter & Whitaker (2009)* discovered a significant relationship between body mass and prey hardness. Prey hardness is highly correlated with body weight, forearm length, total bat length and bat cranial length (*Ayala-Berdon et al., 2023*). Among the bat species, the masseter muscle plays a crucial role in determining the maximum bite force, along with body size (*Senawi et al., 2015*). In term of body size, *H. larvatus* were the largest (FA = 51–67 mm, bite force = 9.40 ±1.73N), followed by *T. melanopogon* (FA = 60–63 mm, body weight = 23–26 g, bite force = 7.78 ±0.91N), *R. coelophyllus* (FA = 41–45 mm, body weight = 6.2–8.6 g), *R. malayanus* (FA = 38–44 mm, body weight = 5–9 g), *R. pusillus* (FA = 37–43 mm, body weight = 3.3–8.0 g) and *R. refulgens* (FA = 33–39 mm, body weight = 3.7–8.2 g, bite force = 1.77 ±0.52 N). Therefore, we can conclude that *H. larvatus* possesses the strongest bite force, followed by *T. melanopogon*, *R. coelophyllus*, *R. malayanus*, *R. pusillus* and *R. refulgens*.

As for the prey hardness, *T. melanopogon* and *H. larvatus* consumed a smaller proportion of hard-bodied prey (*e.g.*, Coleoptera) compared to the four *Rhinolophus* species. During the dry season, *R. coelophyllus* primarily fed on Coleoptera, while *R. pusillus* mainly consumed Lepidoptera. Conversely, in the wet season, *R. malayanus* fed predominantly on Coleoptera, whereas *R. pusillus* consumed Lepidoptera and Hymenoptera. Thus, it can be inferred that larger bat species have a greater capacity to feed on hard-bodied insects (*Ghazali & Dzeverin, 2013*; *Ayala-Berdon et al., 2023*), although this observation is limited to the four *Rhinolophus* species. *Aldridge & Rautenbach (1987)* proposed that larger bats would exhibit a more diverse prey selection, which corresponds to the findings of this study. Notably, *T. melanopogon*, *H. larvatus* and *R. malayanus* consumed a wider range of prey across various insect orders compared to the other bat species.

Since most of the soft-bodied fragments are easily digestible, most of the smaller insect fragments were unidentifiable as they were digested beyond recognition. However, the body fragments of hard-bodied insect orders, such as Coleoptera, Hemiptera and Hymenoptera are heavily chitinized, making them harder to digest, easier to detect and identify in the fecal samples (*Belwood & Fenton, 1976*; *Kunz & Whitaker, 1983*). It is possible that some insect fragments were missed due to their low frequency and being overlooked (*Robinson & Stebbings, 1993*). Additionally, bats may discard certain parts of insects such as legs, wings or antenna before ingestion (*Rabinowitz & Tuttle, 1982*; *Kunz & Whitaker, 1983*; *Robinson & Stebbings, 1993*).

We found that male individuals were generalists, while females were specialists, as they ate specific orders of insects that fit their energy demand in different reproductive stages. Male individuals fed on any insects that were abundant during that season and were observed to fly more actively than female individuals (*Miková et al., 2013*) due to energy demand. In this study, pregnant females were recorded to feed mostly on Coleoptera. Pregnant females consume a specific insect order to obtain a certain nutritional value, consuming food rich in calcium and nitrogen (*Tracy et al., 2006*). If the females cannot obtain food with high nutritional content, they have to consume an abundant variety of insects during pregnancy or lactation (*Barclay, 1994*). Faced with many different nutritional and energetic needs, pregnant females tend to have distinct diets and also foraging strategies compared to non-pregnant females (*Fleming, 1988*). Post-lactating females are generalists and opportunistic feeders due to the low energy demand.

Bat species and insect orders had a significant impact on the percentage of food consumption by bats, as do the factors within female individuals. The findings highlight the importance of bat species and insect order in shaping the dietary preferences of bats. It is noteworthy that different bat species may exhibit distinct food preferences, and even within the same genus, they may consume different insect orders to mitigate competition among and between species (*Salinas-Ramos et al., 2015*; *Roeleke, Johannsen & Voigt, 2018*). Based on the findings, we can conclude that *R. coelophyllus* specialize in feeding on Coleoptera and Diptera, *H. larvatus* feeds on Coleoptera, *R. malayanus* feeds on Hemiptera, *R. pusillus* and *T. melanopogon* feeds on Lepidoptera. Although specific insect pest species consumed by these bats have not been definitively identified, these bat species have the potential to play a crucial role in controlling insect pests in rice fields.

Traditional diet analysis, like done here, necessitates significant time, energy and resources to identify even a single insect fragment. Studies by *Clare et al. (2009)* and *Zeale et al. (2011)* demonstrate that PCR-based methods offer an efficient tool for robust identification of insect prey in the bat fecal samples, even resolving most prey items to the genus or species level. This technique provides valuable insight into the overall ecological relationship of bat diets. Future molecular analysis will be conducted to identify all the insect fragments up to species level, allowing for more detailed exploration of the dietary niche among bat species in the rice fields. This initiative will hopefully enhance our understanding of the dietary relationships of bats populations in rice fields.

## CONCLUSIONS

Gunung Keriang has become an important roosting site for these bat species that forage in the rice fields, emphasizing the hill's significance in bat conservation efforts. This study revealed a distinct diet among bat species, even within the same genus. The bat species and the order of insects consumed play a crucial role in shaping the dietary partitioning of bats. Male and female individuals also exhibited different food preference based on their energy demand, while various reproductive stages influenced the food consumption patterns in females. Larger bats are capable of consuming hard-bodied prey due to their size and maximum bite force. Although we could not identify all fragments up to species level, these bat species have the potential to play a crucial role in controlling insect pests in the rice fields. Future molecular analysis can further confirm the consumed insect pests and provide a better understanding of the dietary relationships of these bat species in rice fields.

## ACKNOWLEDGEMENTS

The authors would like to thank Muda Agricultural Development Authority (MADA) especially Mr. Mohd Ikhwanuddin Khairuddin for providing technical support for this study. Our special thanks also go to the field assistants: Ummu Atiyyah Mohd. Talhah, Aqilah Hanani Mohd. Razani, Muhammad Amir Shuib, Syafi'e Khazali, Mohammad Farhan Rashid, Mohd. Daniel Haiqal Mohd. Zulfikar, Ng Xuanyi, Anis Suraya Kartika Azizan and Nur Azlin Abdul Salim. We also thank Fay Taylor for feedback on earlier versions of this manuscript.

### Funding

This study were funded by Bat Conservation International (BCI) for the Student Research Scholarship (to Nur-Izzati Abdullah) and The Habitat Foundation (to Nurul-Ain Elias). The funders had no role in study design, data collection and analysis, decision to publish, or preparation of the manuscript.

### Grant Disclosures

The following grant information was disclosed by the authors:
Bat Conservation International (BCI) for the Student Research Scholarship (to Nur-Izzati Abdullah).
The Habitat Foundation (to Nurul-Ain Elias).

### Competing Interests

The authors declare there are no competing interests.

### Author Contributions

- Nur-Izzati Abdullah conceived and designed the experiments, performed the experiments, analyzed the data, prepared figures and/or tables, authored or reviewed drafts of the article, and approved the final draft.
- Nurul-Ain Elias conceived and designed the experiments, authored or reviewed drafts of the article, and approved the final draft.
- Nobuhito Ohte conceived and designed the experiments, authored or reviewed drafts of the article, and approved the final draft.
- Christian Vincenot conceived and designed the experiments, authored or reviewed drafts of the article, and approved the final draft.

## Animal Ethics

The following information was supplied relating to ethical approvals (*i.e.,* approving body and any reference numbers):

The Animal Experiment Protocol Kyoto University (In accordance with the provisions of Article 10 of the Regulations on Animal Experimentation at Kyoto University and date of approval: 3rd March 2021 until 31st March 2024).

## Field Study Permissions

The following information was supplied relating to field study approvals (*i.e.*, approving body and any reference numbers):

Universiti Sains Malaysia.

## Data Availability

The raw data is available in the Supplementary File.

## Supplemental Information

Supplemental information for this article can be found online at http://dx.doi.org/10.7717/peerj.16657#supplemental-information.

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
