# Peer review of "Resource partitioning among bat species in Peninsular Malaysia rice fields"

_PeerJ, doi:10.7717/peerj.16657_

## Round 0.1 · original submission · Minor Revisions

Thank you for your submission to PeerJ. Your manuscript has been reviewed by two experts who have provided feedback to improve your manuscript. The reviewers agree that the work is important and presented well. With some minor changes overviewed in their review, the manuscript will be ready for publication. Thank you and I'm looking forward to your next submission

Reviewer 1 ·

Basic reporting

The diversity of insects consumed by bats has been extensively studied in numerous countries, but it remains largely unknown in tropical regions. Nur Izzati and colleagues investigated the diet composition of six bat species in rice fields located in Peninsular Malaysia. They utilized a conventional stereo-microscope to analyze over 1500 fecal samples collected from 301 bats. The researchers provided a comprehensive description of the dietary composition of the bats and discovered that they consumed a wide range of insects, including Coleoptera, Lepidoptera, Diptera, and Hemiptera. The findings also revealed distinct dietary preferences among different bat species, variations between males and females within a species, and differences between the dry and wet seasons. This study offers valuable insights into the feeding habits of bats in tropical rice fields.

The manuscript is clear and well-structured. They provided sufficient field background and literature references. The conclusions are clear and well-stated.

My major comments are as follows:
1. The objective of this study is to evaluate resource partitioning among bats. However, the authors have only reported and compared the dietary composition of the bat species without providing any statistical analysis regarding resource partitioning. It would be helpful if they could further test the dietary niche overlap among the bat species.
2. The authors found that the four Rhinolophus species consumed a high proportion of Coleoptera but a relatively low proportion of Lepidoptera. This finding contrasts with most of the available studies on the diets of Rhinolophus species (e.g., Baroja et al., 2019, PLoS ONE, 14(7), e0219265; Flanders & Jones, 2009, Journal of Mammalogy, 90(4), 888–896; Urtzi et al., 2008, Journal of Mammalogy, 89(2), 493–502). Previous studies have reported that horseshoe bats primarily prey on Lepidoptera and occasionally on other insects such as Diptera and Coleoptera. I suggest the authors explain the reasons that contributed to these different findings. Did the inconsistent findings result from studying bats in different regions or using different methods for diet analysis?
3. Another interesting finding is that T. melanopogon and H. larvatus, which are large-bodied bats, consume a smaller proportion of hard-bodied prey compared to the four Rhinolophus species, which are small-bodied bats. However, the authors state that "Our results align with the findings of Ghazali & Dzeverin (2013) and Ayala-Berdon et al., (2023), which indicate that larger bats are capable of consuming harder insect prey." I am confused about this conclusion. Given their larger body size, T. melanopogon and H. larvatus are expected to consume a larger proportion of hard-bodied prey. They should exercise caution when discussing this finding.

Experimental design

The methods described with sufficient information to replicate. I suggest the authors test the dietary niche overlap among the bat species, which will be helpful to understand the resource partitioning among the bat species.

Validity of the findings

The conclusions are well-stated.

·

Basic reporting

no comment

Experimental design

no comment

Validity of the findings

no comment

Additional comments

This is a very good article in exploring the field of ecosystem services provided by bats, in which is not carried out much in Malaysia. Although the method carried out is quite basic, it still can provide meaningful information to understand the relationship between bats and the environment, especially in agricultural areas. Identifying fragments of bats diet is not an easy task and requires a lot of time and energy. Therefore, this information should be disseminated as widely as possible. Congratulations and good luck.

---

## Round 0.2 · Minor Revisions

Thank you for revising your work and resubmitting your manuscript. The authors have done an excellent job addressing the comments in the rebuttal letter, but have not always directly assessed the comments in the manuscript itself. For example, R1 asked for an understanding of dietary niche partitioning. It is okay that the authors believe this to be outside the scope of the current manuscript, but they need to address the comment directly in the text because it will no doubt be a thought for many of the readers of this paper. Why was the dietary niche not included? is it a future direction? etc. There was also a comment by R2 which you answered in red - this or a version of this should be directly in the manuscript.

R2 commented about the use of acronyms throughout - I would have to agree, that acronyms are confusing to readers who are not already familiar with them. Where possible, please remove acronyms and replace them with the full name (like MADA which is a location-specific department and will confuse readers from other regions of the world.

---

## Round 0.3 · accepted · Accept

Thank you for making those minor edits to the manuscript. I now believe that the manuscript is acceptable for publication, congratulations!